# Does Farmers' Lottery Participation Affect Technical Efficiency of Banana Production in Rural China?

Mingze Wu [1,2] and Yueji Zhu [2,*]

1   College of Economics and Management, South China Agricultural University, Guangzhou 510642, China
2   Management School, Hainan University, Haikou 570228, China
*   Correspondence: yzhu@hainanu.edu.cn

**Abstract:** Increasing the agricultural technical efficiency is crucial to poverty reduction for smallholder farmers in developing countries. This study attempts to examine the impact of farmers' lottery participation on technical efficiency from the perspective of the farmers' revealed risk preferences, and to explore the influence mechanism between lottery participation and technical efficiency, based on the primary data collected from banana farmers in rural China. We used data envelopment analysis (DEA) to measure the combined technical efficiency of the farmers and constructed an endogenous switching regressions (ESR) model to analyse the impact of lottery participation on the technical efficiency of banana farms. A quantile regression model was used to analyse the heterogeneous effects under the different levels of the farmers' technical efficiency. Then, a mediation model was employed to clarify the influence mechanism of lottery participation on technical efficiency in terms of two pathways: the adoption of biopesticide and the soil improvement technique. The empirical results show that farmers' lottery participation revealed their risk preferences and several factors affected banana farmers' participation in the lottery. Specifically, male farmers are more likely to participated in the lottery than female; farmers' working hours negatively affected their lottery participation; and the use of a smartphone significantly increased the likelihood of farmers buying lottery tickets. We also found that farmers who participate in the lottery have higher technical efficiency in banana production, and the average treatment effect of lottery participation on the technical efficiency was 21.5%, indicating that the farmers with revealed risk preferences can significantly promote technical efficiency. The effect of risk preferences on economic performance is more significant for farmers at the middle technical efficiency level. The explanation is that the adoption of new technologies (e.g., biopesticides) played a mediating effect between farmers' lottery participation and their technical efficiency. New technologies are more likely to be adopted by farmers who participate in the lottery, resulting in higher technical efficiency. Therefore, policymakers and stakeholders can better design technology extension programs according to the different attitudes of the target farmers towards risks in developing regions.

**Keywords:** revealed risk preferences; technical efficiency; technology adoption; mediating effect; banana farmers





## 1. Introduction

Growing recognition has been attached to the important role of improving technical efficiency in poverty reduction for smallholder farmers in developing countries [1–3]. The productivity of small-scale farms is low, and some researchers suggest that poverty alleviation in extremely poor areas is more likely to be achieved through urbanisation and a radical transformation of the agricultural sector [3,4]. For a developing country with a large population, such as China, improving the efficiency of crop production is necessary in order to achieve sustainable agricultural development and food security. With merely 7% of the world's arable land to feed 21% of the world's population, it is particularly urgent to enhance the agriculture technical efficiency in China [5].

How to effectively improve the efficiency of agricultural production in developing countries has long been a concern for many scholars [6–8]. From the perspective of resource allocation, Schultz [9] deemed that the fragmentation of land resources has led to a misallocation of agricultural production factors. This has hindered farmers' adoption of technologies and the construction of farm infrastructure, and increased the technically inefficient component, therefore reducing the efficiency of inputs. Similarly, urban expansion leads to land fragmentation, which increases the transaction cost of investments and affects production efficiency [10–12]. The inequitable distribution of arable land may also lead to large differences in the efficiency of agricultural production [13]. The application of new technologies and management practices in agricultural production promotes the rational allocation of resources and reduces production costs. As a result, crop yields and farmer incomes have grown and agricultural technical efficiency has increased. It can be challenging to increase the productivity of farms if smallholder farmers lack the skills to recognise and employ new technologies [14,15].

The agricultural production of small-scale farms in developing countries is highly vulnerable to natural disasters and market price fluctuations [16,17]. The natural and market risks threaten the livelihoods of smallholder farmers. In existing studies, the theory of planned behaviour (TPB) has been widely used to understand farmers' intentions or behaviour in agricultural production [18,19]. Under risks and uncertainties, smallholder farmers often make decisions based on their prior farming experiences. In order to effectively prevent shocks to their livelihoods and to stabilise crop yields, farmers could adopt ex-ante comprehensive management strategies, such as crop diversification, agricultural insurance, etc. [20–22]. However, it cannot be overlooked that the adoption of coping strategies is inseparable from the attitudes of smallholders towards risks. Risk preference refers to farmers' attitudes and decision-making behaviour under risks in agricultural production. According to previous studies, farmers' risk preferences have three categories: risk-seeking, risk-neutral and risk-averse [23–25]. The existing literature also suggests that the adoption of new technologies is often associated with farmers' risk preferences [26,27]. Risk-averse farmers are more likely to follow traditional practices and technologies, while risk-seeking farmers are more likely to embrace new crop varieties and technologies and benefit from them [24,28,29].

The measurement of individual's risk preference has been extensively discussed, particularly in the following three aspects. The first is a self-report method based on Likert scales, but it is susceptible to the adverse effects of respondents' perceptions and attitudes [30]. The second is the contextualised game experiments, which can easily distinguish respondents' levels of risk preference; the problem is that the rationality of the game design itself can easily be questioned [31]. The third is the econometric estimates based on collected data, such as coding strategy design [32] and positive mathematical programming [33], but the experimental design is extensive, time-consuming and difficult for respondents to understand. In contrast to the above methods, this study uses farmers' observed choices regarding lottery participation as the proxy in order to reveal their actual risk preferences. Participation in lottery is a specific act that reveals risk preferences, and risk-seekers tend to be more enthusiastic about the lottery [34–36]. It can help us avoid the possible biased classification caused by the methods of measurement on risk preference. However, if one is addicted to the lottery, it may cause a series of serious social problems [37,38].

This work has the following two research objectives. The primary objective of this study is to examine the impact of farmers' lottery participation on their technical efficiency. Generally, lottery participation can reveal farmers' risk preferences; risk-seeking farmers may be more inclined to use new technologies or practices to increase the technical efficiency. However, it also potentially has a negative impact on agricultural production. For example, farmers may spend too much time and energy on the lottery, and it may reduce their inputs into agriculture. Thus, it is worth an empirical exploration. The second objective of this research is to detect the mediating factors between the two variables to ascertain whether farmers' lottery participation affects their technical efficiency. This study suggests two

possible pathways. The first path is that farmers are addicted to lotteries, reporting a passive attitude towards farming, thus inhibiting their technical efficiency. Another path is that farmers with lottery participation are risk-seekers, and prefer to adopt new technologies in agriculture, thereby increasing their technical efficiency. Thus, the adoption of new technology can be a reasonable mediating factor between farmers' lottery participation and their agricultural technical efficiency. This paper tests this using a mediation model based on the data collected from banana farmers in rural China.

This study may contribute to the existing literature in two aspects. First, we use the observed action of lottery participation as the revealed risk preference of smallholder farmers, rather than a paper test of risk preference, to examine the association between farmers' lottery participation and technical efficiency. The association has rarely been discussed in previous studies. Second, farmers' decisions of lottery participation are not random events, and depend on a series of observable factors and unobservable factors. To avoid the biased estimation caused by the possible endogeneity problem, we employ an endogenous switching regressions (ESR) model to estimate the impact of farmers' lottery participation on their technical efficiency.

The remainder of this paper is organised as follows: Section 2 presents the estimation strategy; Section 3 introduces the study area and data collection for this paper, and presents the descriptive statistics of the variables; the empirical results and discussion are given in Section 4. The final section presents the conclusion and policy implications.

## 2. Estimation Strategy

### 2.1. DEA-CCR Model

The Data Envelopment Analysis (DEA) model is used to estimate the technical efficiency of the sample farmers. This study selects the CCR form of the DEA model [39], which evaluates the efficiency of a decision-making unit (DMU) with the assumption of constant returns to scale (CRS) as a prerequisite. Given the output as the constant, the input-oriented model is as follows:

$$max \frac{\sum_{r=1}^{s} u_r y_{r0}}{\sum_{i=1}^{m} v_i x_{i0}} \tag{1}$$

$$s.t. \begin{cases} \frac{\sum_{r=1}^{s} u_r y_{rj}}{\sum_{i=1}^{m} v_i x_{ij}} \leq 1, j \in [1, n] \\ u_r, v_i \geq \varepsilon, r \in [1, s], i \in [1, m] \end{cases} \tag{2}$$

A linear programming form equivalent to Equation (1) is obtained using the Charnes-Cooper transformation [40], as follows:

$$max \sum_{r=1}^{s} u_r y_{r0} \tag{3}$$

$$s.t. \begin{cases} \sum_{r=1}^{s} u_r y_{rj} - \sum_{i=1}^{m} v_i x_{ij} \leq 0, j \in [1, n] \\ \sum_{i=1}^{m} v_i x_{i0} = 1 \\ u_r, v_i \geq \varepsilon, r \in [1, s], i \in [1, m] \end{cases} \tag{4}$$

where $u_r, v_i$ are the vectors of the *r*-th output and the *i*-th input, respectively. $\varepsilon$ is a non-Archimedean infinitesimal. Following the existing studies [41–43], the output variable is the total banana production in 2020 and the input variables are land, capital, labour and intermediate input (Table 1).

**Table 1.** Description of input and output indicators.

| Indicator | Variables | Description of Indicators (Units) |
|---|---|---|
| Output | Banana production | Total banana production in 2020 (Kg) |
| | Land | Area of bananas grown in 2020 (Mu [a]) |
| | Capital | Farmers' expenditure on seedlings, machinery, irrigation, hired labour (CNY) |
| Input | Labour | Number of household labours multiplied by the actual number of working days in the farm (Days) |
| | Intermediate input | Farmers' expenditure on pesticides, fertilisers and other inputs (CNY) |

Note: [a] Mu is a Chinese unit of measurement, with 1 Mu = 1/15 hectares.

### 2.2. The Endogenous Switching Regression Model

To tackle the issue of selection bias in the sample, the propensity score matching (PSM) model and inverse probability-weighted with a regression adjustment (IPWRA) estimator have been used in the empirical studies [20,44]. In contrast, the ESR model has three advantages: first, it takes into account the endogeneity problem caused by both the observable and unobservable factors that may affect farmers' choices on whether or not to participate in the lottery. Second, the outcome equation for banana farmers who participated in the lottery were regressed separately from those who did not, so that two technical efficiency equations could be estimated jointly in order to better fit the effect of each variable on the technical efficiency. Finally, the problem of missing valid information can be better avoided by using a full information maximum likelihood estimation [44,45]. Thus, the ESR model was used to estimate the effect of banana farmers' lottery participation on their technical efficiency in this study. The ESR model jointly estimates the following two equations:

The first is a selection equation to indicate the banana farmers' lottery participation:

$$lottery_i = \alpha Z_i + \mu_i, lottery_i = \begin{cases} 1, lottery_i > 0 \\ 0, lottery_i \leq 0 \end{cases} \tag{5}$$

The second is an outcome equation of the treatment group for the technical efficiency of banana farmers with lottery participation:

$$productivity_1 = \beta_1 X_i + v_{1i}, \text{if } lottery_i = 1 \tag{6}$$

The other outcome equation is of the control group for the technical efficiency of banana farmers who did not participate in the lottery:

$$productivity_0 = \beta_2 X_i + v_{2i}, \text{if } lottery_i = 0 \tag{7}$$

where $lottery_i$ in Equation (5) denotes the binary selection variable to indicate whether a banana farmer participated in the lottery or not, $Z_i$ is a vector of variables that may affect the farmer's decision on lottery participation, including the characteristics of the farmer, farm household and social capital, $\mu_i$ is a random error term. $productivity_1$ and $productivity_0$ in Equations (6) and (7) denote the agricultural technical efficiency of the two groups of participants and non-participants of the lottery, respectively. $X_i$ is a vector of a series of factors that may affect the technical efficiency of banana farmers. $v_i$ is a random error term.

The three error terms $\mu_i$, $v_{1i}$ and $v_{2i}$ in Equations (5)–(7) are assumed to have a triumvirate normal distribution with zero mean, and the variance-covariance structure is [46]:

$$cov(\mu_i, v_{1i}, v_{2i}) \begin{bmatrix} \sigma_{\mu_i}^2 & \sigma_{\mu v_{1i}} & \sigma_{\mu v_{2i}} \\ \sigma_{v_{1i}\mu} & \sigma_{v_{1i}}^2 & \cdot \\ \sigma_{v_{2i}\mu} & \cdot & \sigma_{v_{2i}}^2 \end{bmatrix} \tag{8}$$

where $\sigma_{\mu_i}^2$, $\sigma_{v_{1i}}^2$ and $\sigma_{v_{2i}}^2$ are the variances of the error terms in the selection Equation (5) and the outcome Equations (6) and (7), respectively. $\sigma_{\mu v_{1i}}$ denotes the covariance of $\mu_i$ and $v_{1i}$, and $\sigma_{\mu v_{2i}}$ denotes the covariance of $\mu_i$ and $v_{2i}$.

To ensure that the ESR model can be identified, we include the "attitudes of family members" as an instrumental variable. The regression results show that the instrumental variable significantly influences farmers' lottery participation, while not directly affecting the technical efficiency, suggesting that the instrumental variable is appropriate in this model.

The ESR model estimates the effects of lottery participation on technical efficiency separately for the two groups of farmers. Based on the counterfactual framework, it is possible to measure the overall impact of farmers' lottery participation on the technical efficiency of the entire sample of farmers by calculating the average treatment effect (ATE) [47]:

$$ATE = E(productivity_i|lottery_i = 1) - E(productivity_i|lottery_i = 0) \tag{9}$$

where $E(productivity_i|lottery_i = 1)$ in Equation (9) denotes the expected average technical efficiency when all of the sample farmers participated in the lottery, and $E(productivity_i|lottery_i = 0)$ denotes the expected average technical efficiency when all of the sample farmers did not participate in the lottery. The ATE calculated by Equation (9) has controlled the self-selection bias.

### 2.3. The Quantile Regression Model

This study introduced a quantile regression model for estimation, aimed to describe the relationship between farmers' lottery participation and technical efficiency more comprehensively. We chose several representative quartiles (i.e., 10th, 30th, 50th, 70th and 90th quantiles). The bootstrap method is used to estimate the standard error in the quantile regression. Koenker and Bassett [48] proposed a quantile regression model to overcome the limitations of the OLS method. The quantile regression model takes the weighted average of the absolute values of the residuals as the objective function for minimisation, and it is less susceptible to outliers and can provide more accurate information about the conditional distribution [49]. The quantile regression can use the full sample data to estimate the parameters of the different quantiles, in contrast to conventional segmental regression methods [50,51]. The empirical form of the quantile regression model is given as:

$$I_j = \theta_\tau x_j' + \omega_{\tau j}, 0 < \tau < 1 \tag{10}$$

$$Quant_\tau(I_j|x_j) = \theta_\tau x_j \tag{11}$$

where $x_j'$ denotes the vector of the farmers' lottery participation, $I_j$ denotes the farmers' technical efficiency, $\omega_{\tau j}$ is a random error term. $Quant_\tau(I_j|x_j)$ is the $\tau$th quantile of the farmers' technical efficiency. $\theta_\tau$ is the coefficient of the $\tau$th quantile, and its regression estimator $\hat{\theta}_\tau$ can be a solution of the following formula:

$$min\sum_{j:I_j \geq \theta_\tau x_j'}^{n} \tau \mid I_j - \theta_\tau x_j' \mid + \sum_{j:I_j < \theta_\tau x_j'}^{n} (1-\tau) \mid I_j - \theta_\tau x_j' \mid \tag{12}$$

when $\tau$ is equal to different values, it can obtain different parameter estimates. The median regression is a special case of quantile regression, under the condition that $\tau$ is equal to 0.5.

### 2.4. Mediation Model

The theoretical basis of the mediation model is rooted in social psychology, where researchers often seek to understand the mechanisms of how one factor influences another [29]. The mediation model makes the assumption that the mediator partially or completely explains the link between the independent and dependent variable. To verify the mediating role of technology adoption between banana farmers' lottery participation

and the technical efficiency of their farms, this study follows a stepwise regression method from Baron and Kenny [52]:

$$\begin{cases} productivity_i = c_0 + c_1 lottery_i + c_2 X_i + \epsilon_i \\ M_i = a_0 + a_1 lottery_i + a_2 X_i + \delta_i \\ productivity_i = b_0 + b_1 lottery_i + b_2 M_i + b_3 X_i + \varphi_i \end{cases} \tag{13}$$

where $M_i$ is the mediating variable; $a_0, b_0, c_0$ are the constant terms; $a_1, b_1, c_1, a_2, b_2, c_2$ are the parameters to be estimated; and $\epsilon_i, \delta_i, \varphi_i$ are the error terms. $c_1$ is the overall effect of the $lottery_i$ on the $productivity_i$; $b_1$ is the direct effect of the $lottery_i$ on the $productivity_i$ after controlling for the effect of the $M_i$; the mediating effect has the following relationship with the overall and direct effects:

$$c_1 = b_1 + a_1 b_2 \tag{14}$$

If the coefficient on $c_1$ is significant, it is considered as the mediating effect, otherwise it is considered to be a masking effect; if both coefficients $a_1$ and $b_2$ are significant, it is considered as an indirect effect, otherwise a Sobel test or Bootstrap test is required. Next, the signs of $a_1 b_2$ and $b_1$ can be compared. When both have the same sign, they are considered to have a partial mediating effect and show the ratio of $a_1 b_2 / b_1$; if the signs of the two are different, they are considered to have a masking effect and show the ratio of $|a_1 b_2 / b_1|$.

## 3. Data Description

### 3.1. Study Area and Data Collection

The data for this study were collected through a farmer household survey in 2021. The sample farmers are randomly selected from the major banana production areas in Hainan province, which is a tropical region in the south of China. According to the China National Banana Industry Technology System (CNBITS), the total production of banana in China was 5.348 million tons in 2020, while Hainan province ranks as the third largest production area in China, with a total production of 1.084 million tons.

A multistage sampling method was used for the data collection. Firstly, four counties in western Hainan Province—Chengmai, Lingao, Changjiang and Ledong (see Figure 1), which are the main banana growing areas in the province—were deliberately selected as the sample sites for the study. According to the Hainan Provincial Bureau of Statistics, the area of bananas harvested in the aforementioned four counties accounted for 63.5% of the province's total in 2019; therefore, the four sample counties are of considerable relevance and representativeness. Secondly, two to four towns were randomly selected from each sample county. In the study area, a town actually consists of serval villages; three to five villages were randomly selected from each town. To ensure the representativeness of the sample and the reliability of the statistical analysis, no more than 20 banana farmers were randomly selected from each village as survey respondents. The face-to-face interviews with farmers were conducted to ensure the quality of the collected data. The survey questionnaire covers the characteristics of the banana farmers, their household characteristics, social characteristics, the inputs and outputs in the banana production and the farmers' lottery participations.

The data were collected by a farmer household survey conducted by trained postgraduate students. We conducted a pilot survey in Chengmai County of Hainan province and revised the questionnaire accordingly. To avoid the duplication of the farmer household data, team members checked the households' identification information before the face-to-face interview. Finally, 422 valid sample farmers were collected for this study. The sample distribution is given in Table 2.

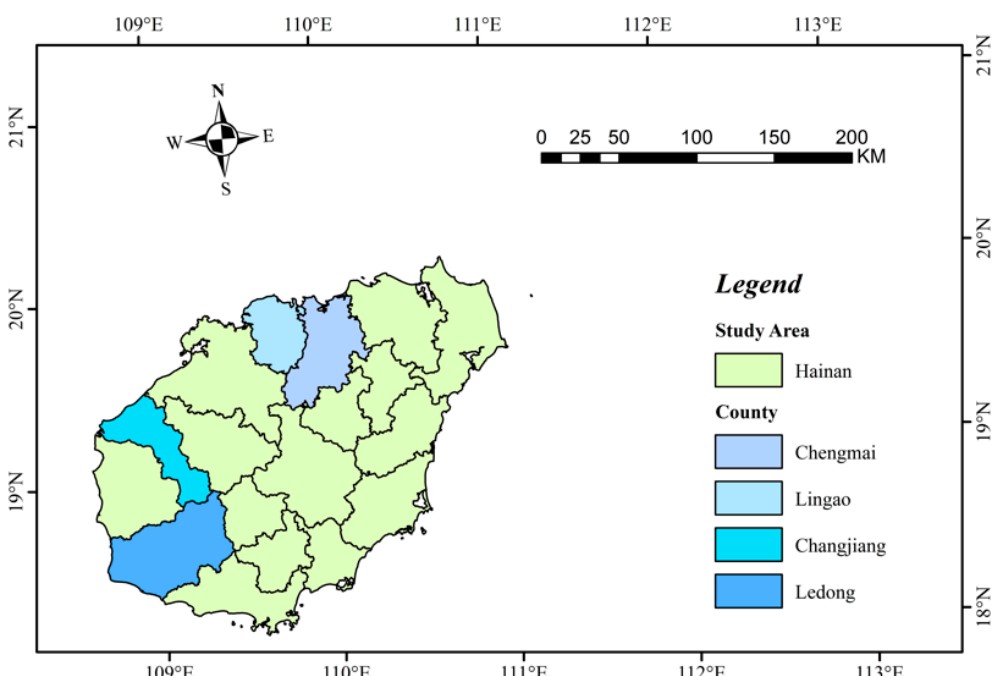

**Figure 1.** Map of study area.

**Table 2.** Sample distribution.

| County | Town | Observations | Percentage (%) |
|---|---|---|---|
| Chengmai | Dafeng, Jinjiang | 100 | 23.70 |
| Lingao | Huangtong, Dongying, Diaolou, Lincheng | 101 | 23.93 |
| Changjiang | Changhua, Shiyuetian, Shilu | 105 | 24.88 |
| Ledong | Liguo, Huangliu | 116 | 27.49 |

*3.2. Descriptive Statistics*

3.2.1. Variables and Measurement

Table 3 presents the definitions of the variables and the descriptive statistics. The dependent variable is the technical efficiency of the banana farmers. Based on multiple inputs (land, capital, labour and intermediate input) and a single output (banana yield) of the farmers, the technical efficiency can be calculated by the DEA-CCR model.

The technical efficiency of the sample farmers averaged 0.55, indicating that the production factors are not used with high efficiency by these banana farmers. The farmers' lottery participation is the core explanatory variable in this paper. The lotteries issued in China are mainly the "welfare lottery" and "sports lottery", and their purpose is to raise funds for social welfare and sports [53]. Among the sample farmers, 249 farmers participated in lotteries, while 173 farmers did not.

Following prior studies [20,54,55], the characteristics of the individuals, farm households and social capital are included in this study as control variables. The individual characteristics are captured by age, gender, education, farming experience, health condition and working hours; the household characteristics include family labour, loans, land area and internet and smartphone use, while the social characteristics include off-farm work, cooperative member, tie to extension workers and agricultural retailers.

The indicators "biopesticides" and "soil improvement" are used as the mediating variables in this study. Biopesticides are new and safer for humans and animals than chemical pesticides. The active ingredients are fully derived from natural ecosystems and have a less negative impact on the environment. The use of soil improvement techniques can improve the soil structure, reduce saline hazards, increase the use efficiency of water

and fertiliser and promote crop yields. The proportions of farmers who have adopted biopesticides and soil improvement were 39.3% and 68.5%, respectively.

**Table 3.** Variable definitions and descriptive statistics.

| Variables | Definition | Mean | Std. Dev. |
|---|---|---|---|
| *Explained variable* | | | |
| Technical efficiency | Actual values obtained from measurements using the DEA model | 0.546 | 0.254 |
| *Explanatory variable* | | | |
| Lottery participation | Have you participated in the lottery in the last year? 1 = yes, 0 = otherwise | 0.590 | 0.492 |
| *Control variables* | | | |
| Age | Farmers' age (years) | 47.137 | 11.039 |
| Gender | 1 = male, 0 = female | 0.730 | 0.445 |
| Education | Farmer's education level (years) | 8.422 | 3.209 |
| Farming experience | Experience in agriculture (years) | 24.806 | 12.521 |
| Health condition | Farmer's health condition: 1 = very bad health, 5 = very good health | 3.637 | 1.098 |
| Working hours | Average daily working hours a day of the respondents | 9.314 | 2.550 |
| Family labour | Number of family members engaged in banana production | 2.602 | 1.281 |
| Loans | Does the household have a loan from the bank? 1 = Yes, 0 = No | 0.408 | 0.492 |
| Land area | cropping area in mu | 21.862 | 35.435 |
| Internet use | 1 = if farmer uses the WI-FI, 0 = otherwise | 0.711 | 0.454 |
| Smartphone | Do you use your smartphone to access agricultural information? 1 = yes, 0 = otherwise | 0.372 | 0.484 |
| Off-farm work | 1 = the farmer was engaged in off-farm work, 0 = otherwise | 0.467 | 0.500 |
| Cooperative member | 1 = cooperative member, 0 = otherwise | 0.078 | 0.269 |
| Tie to extension workers | Degree of contact with the extension workers? 1 = no contact, 5 = extremely close contact | 1.607 | 1.042 |
| Tie to agricultural retailers | Degree of contact with the agricultural retailers? 1 = no contact, 5 = extremely close contact | 3.700 | 1.091 |
| *mediator variables* | | | |
| Biopesticides | Use of biopesticides? 1 = yes, 0 = otherwise | 0.393 | 0.489 |
| Soil improvement | Adopt soil improvement techniques? 1 = yes, 0 = otherwise | 0.685 | 0.465 |
| *Instrumental variable* | | | |
| Attitude of family members | Attitude of family members towards participation in the lottery? 1 = against, 2 = neutral, 3 = support | 1.765 | 0.592 |

The attitude of family members towards the lottery may influence farmers' lottery participation, but it does not directly affect the agricultural technical efficiency. Therefore, it is possible to use "attitude of family members" as an instrumental variable in the subsequent estimation equations. Table 3 shows that the family members did not support the lottery participation.

### 3.2.2. Comparison between Participants and Non-Participants

Table 4 presents the mean value and difference (*t*-test) of each variable for the two groups of farmers, namely the participants and non-participants of the lottery. Firstly, the technical efficiency of the farmers who participated in the lottery was higher than that of those who did not. According to the *t*-test results, there is a statistically significant difference between the two groups of farmers in terms of their technical efficiency. Secondly, the participants were more likely to be male. The participants were more likely to use a smartphone to access agricultural information and they were also more inclined to use biopesticides in agriculture. However, the participants had fewer working hours than

the non-participants. Among the social characteristics, the non-participants had a closer relationship with agricultural retailers. This is a preliminary description of the two groups of farmers, and a more solid estimation of the impact of farmers' lottery participation on the technical efficiency is given in the following section.

**Table 4.** Difference of characteristics between participants and non-participants.

| Variables | Treated Group | Control Group | *p*-Value |
|---|---|---|---|
| Technical efficiency | 0.564 | 0.521 | 0.088 |
| Age | 46.811 | 47.607 | 0.467 |
| Gender | 0.775 | 0.665 | 0.012 |
| Education | 8.538 | 8.254 | 0.372 |
| Farming experience | 24.165 | 25.728 | 0.207 |
| Health condition | 3.683 | 3.572 | 0.310 |
| Working hours | 9.094 | 9.630 | 0.034 |
| Family labour | 2.538 | 2.694 | 0.221 |
| Loans | 0.394 | 0.428 | 0.483 |
| Land area | 20.879 | 23.277 | 0.495 |
| Internet use | 0.735 | 0.676 | 0.192 |
| Smartphone | 0.410 | 0.318 | 0.055 |
| Off-farm work | 0.486 | 0.439 | 0.346 |
| Cooperative member | 0.072 | 0.087 | 0.589 |
| Tie to extension workers | 1.618 | 1.590 | 0.780 |
| Tie to agricultural retailers | 3.618 | 3.815 | 0.069 |
| Biopesticides | 0.430 | 0.341 | 0.067 |
| Soil improvement | 0.695 | 0.671 | 0.599 |
| Observations | 249 | 173 | |

## 4. Results and Discussion

### 4.1. Impact of Farmers' Lottery Participation on Technical Efficiency

The impact of farmers' lottery participation on their agricultural technical efficiency is estimated using the ESR model, and the results are presented in Table 5. In the lower panel of Table 5, the Wald test rejects the null hypothesis that the selection equations and outcome equations are independent of each other at the 1% level. Moreover, both $ln\sigma_1$ and $ln\sigma_0$, which reflect the correlation between $\mu_i$ and $v_i$, are negative and significantly different from zero, indicating that some unobservable factors influenced both the farmers' lottery choices and their technical efficiency [44]. The estimates of $\rho0$ are not significantly correlated, while $\rho1$ are negative correlation coefficients. This indicates a negative selection bias, implying that the impact of lottery participation on technical efficiency could be underestimated if the selection bias of the sample farmers is neglected [56,57]. Therefore, the estimation is more accurate using the ESR model than the OLS model.

The first stage of the ESR model estimated the determinants of the farmers' lottery participation, and the results are presented in the third column of Table 5. Male farmers, smartphone use, and family members' attitude towards the lottery showed a statistically significant and positively association with the farmers' lottery participation. Specifically, men are more inclined to participated in lotteries than women, a finding that is consistent with the findings in previous studies [58]. Generally, men have a stronger preference for adventure than women. Smartphone usage can promote farmers' lottery participation; this finding is somehow consistent with Törrönen et al. [59]. The coefficient of the attitude of family members towards lottery is positive and significant, suggesting that if family members have a supportive attitude towards the lottery, it is more likely the farmer participated in it. Existing evidence also showed that children are involved when their parents are addicted to lotteries [60]. It should be noted that the selection equation in the ESR model accounts for unobserved heterogeneity that could bias the treatment effect of lottery participation on technical efficiency. For this reason, the selection equation needs to include a valid instrument, which should be excluded in the outcome equation [44]. In contrast,

working hours, family labour, and tie to agricultural retailers negatively influenced the farmers' lottery participation. Picchio et al. [61] suggested that winning the lottery prize could reduce people's working hours, while our study shows that the probability of farmers' lottery participation decreases significantly as their average working hours and family labour increase. Agricultural retailers are the important node for transmitting the market information among farmers in rural China. The tie to agricultural retailers shows a negative and statistically significant association with the farmers' lottery participation, suggesting that farmers with closer interaction with agricultural retailers are less likely to participate in the lottery.

**Table 5.** The ESR estimation results.

| Category | Variables | Selection | Outcome: Technical Efficiency | |
| --- | --- | --- | --- | --- |
| | | Lottery Participation | Treated Group | Control Group |
| Individual | Age | −0.004 (0.011) | 0.003 (0.003) | −0.006 ** (0.003) |
| | Gender | 0.565 *** (0.161) | 0.032 (0.048) | 0.185 *** (0.050) |
| | Education | −0.017 (0.022) | 0.001 (0.006) | −0.013 ** (0.006) |
| | Farming experience | −0.002 (0.010) | −0.004 (0.003) | 0.004 (0.003) |
| | Health condition | −0.017 (0.064) | −0.010 (0.018) | 0.009 (0.017) |
| | Working hours | −0.052 ** (0.025) | 0.014 * (0.007) | 0.007 (0.008) |
| | Family labour | −0.111 ** (0.051) | 0.011 (0.015) | 0.027 * (0.015) |
| Household | Loans | −0.204 (0.133) | −0.004 (0.037) | −0.056 (0.036) |
| | Land area | 0.000 (0.002) | 0.001 * (0.001) | 0.001 *** (0.001) |
| | Internet use | 0.199 (0.141) | −0.089 ** (0.042) | 0.021 (0.040) |
| | Smartphone | 0.285 ** (0.145) | −0.076 * (0.039) | 0.012 (0.045) |
| | Off-farm work | 0.046 (0.133) | 0.034 (0.037) | −0.042 (0.037) |
| Social | Cooperative member | −0.117 (0.235) | 0.006 (0.068) | 0.093 (0.062) |
| | Tie to extension workers | 0.022 (0.063) | −0.014 (0.018) | 0.017 (0.017) |
| | Tie to agricultural retailers | −0.139 ** (0.062) | 0.051 *** (0.017) | −0.007 (0.020) |
| IV | Attitudes of family members | 0.450 *** (0.105) | - | - |
| | _cons | 0.600 (0.686) | 0.430 (0.176) | 0.509 (0.232) |
| | lnσ1 | - | −1.194 *** (0.086) | - |
| | ρ1 | - | −1.310 *** (0.314) | - |
| | lnσ0 | - | - | −1.511 *** (0.055) |
| | ρ0 | - | - | −0.049 (0.401) |
| | likelihood ratio test | | 6.75 ** | |
| | Wald test | | 57.46 *** | |
| | Observations | | 422 | |

Note: Standard errors in parentheses. Significance level: *** $p < 0.01$, ** $p < 0.05$, * $p < 0.1$.

The ESR estimation results for the outcome equation of the participants and non-participants are shown in the fourth and fifth columns of Table 5. The technical efficiency of the farmers in the two groups is affected by different factors. For the treated group, working hours shows a positive sign, with a significance level at 10%, indicating that the working hours in the farm can increase the technical efficiency of the banana farmers who participated in the lottery because the more time banana farmers devote to labour production, the more they pay attention to the quality of the agricultural production. However, this finding contradicts the result reported by Qing et al. [62]. The coefficient of land area is positive and statistically significant, suggesting that technical efficiency can be increased if farmers cultivate larger land. The coefficients of internet use and smartphone respectively show a negative and statistically significant impact on technical efficiency of lottery participants. The use of internet technology may increase farmers' income and reduce their investment in agricultural production [63,64]. In the field research, we learned that some farmers put all of their agricultural profits into the lottery, rather than into agricultural production. A tie to agricultural retailers shows a positive sign, with a significance level at 1%, indicating that the closer the contact between farmers and

agricultural retailers, the higher the technical efficiency. The farmers in the treated group were more likely to consult agricultural retailers about banana management. They can also learn the new techniques of pesticides and fertiliser, thus contributing to their increased technical efficiency [65,66].

In terms of the control group, a significant and negative correlation between the farmers' age and agricultural technical efficiency is presented in the fifth column. This means that older farmers may have lower technical efficiency. The older farmers usually tend to be risk-averse and may be not interested in new agricultural technologies. Conservative risk attitudes render the elder famers less productive than the younger farmers [67]. The male-headed household is significantly more productive than the female-headed household at the 1% statistical level. Agarwal [68] argued that the male has more experience in farm management than the female, and can optimise the allocation of agricultural resources, thereby improving their technical efficiency. Generally, a higher education level can help farmers improve their crop yields and efficiency [69], while this study presents an opposite result in the control group. The banana farmers with higher education have lower technical efficiency because they are more likely to engage in off-farm work and have less investment in farming. Family labour is positively correlated with technical efficiency in the control group, indicating that the more members of the household engaged in agriculture, the higher the technical efficiency of their farms. In addition, the scale effect of the labour facilitates the optimal allocation of production materials and provides more economic benefits [70].

### 4.2. Average Treatment Effect of Farmers' Lottery Participation on Technical Efficiency

The average treatment effect of lottery participation on the banana farmers' technical efficiency is measured based on Equation (9), and the results are presented in Table 6. The technical efficiency of the participants and non-participants is 0.732 and 0.517, respectively. The average treatment effect of lottery participation on the banana farmers' technical efficiency is positive, with significance at the 1% level. This implies that the farmers who participated in the lottery have a 41.59% higher technical efficiency than those who did not, controlling for observable and unobservable factors. Evidently, farmers with revealed risk preferences have higher agricultural technical efficiency. The positive correlation between the farmers' risk preference and technical efficiency was supported by previous studies [25,71].

**Table 6.** Average treatment effect estimation results.

| Technical Efficiency | Whether Lottery Participation | | ATE | *t*-Value | Standard Errors | Change (%) |
|---|---|---|---|---|---|---|
| | Treated Group | Control Group | | | | |
| | 0.732 | 0.517 | 0.215 *** | 28.136 | 0.008 | 41.59 |

Note: *** $p < 0.01$, Change (%) = [(0.732 − 0.517)/0.517] × 100%.

### 4.3. Heterogeneous Analysis

The estimated results of the quantile regressions are provided in Table 7. Specifically, the coefficient of lottery participation on technical efficiency at the 50th quantile is the biggest (0.075), while the coefficient at the 30th quantile is 0.066. Both are statistically significant, suggesting that the effect of risk preferences on economic performance is more significant for farmers with middle levels of technical efficiency. The correlation between lottery participation and technical efficiency is not statistically significant at the 70th and 90th quantiles. Farmers with high productivity may use more advanced production technologies and spend their time on farming rather than lotteries [62,70]. The results imply that the revealed risk preferences have heterogeneous effects across the different levels of the farmers' technical efficiency.

**Table 7.** Quantile regression model estimation results.

| Variables | 10th Quant | 30th Quant | 50th Quant | 70th Quant | 90th Quant |
|---|---|---|---|---|---|
| Lottery participation | 0.032 | 0.066 * | 0.075 ** | 0.003 | 0.047 |
| | (0.046) | (0.040) | (0.035) | (0.039) | (0.040) |
| Control variables | | | Controlled | | |
| _cons | 0.203 | −0.023 | 0.157 | 0.485 ** | 0.951 *** |
| | (0.210) | (0.222) | (0.180) | (0.198) | (0.211) |
| Observations | | | 422 | | |

Note: Bootstrap standard errors in parentheses. Significance level: *** $p < 0.01$, ** $p < 0.05$, * $p < 0.1$.

To illustrate the variation of the estimated coefficients under the different quantiles, we report the trends of the coefficients for significant variables in Figure 2. The shape of the curves basically confirms the change of the estimated coefficients in Table 7.

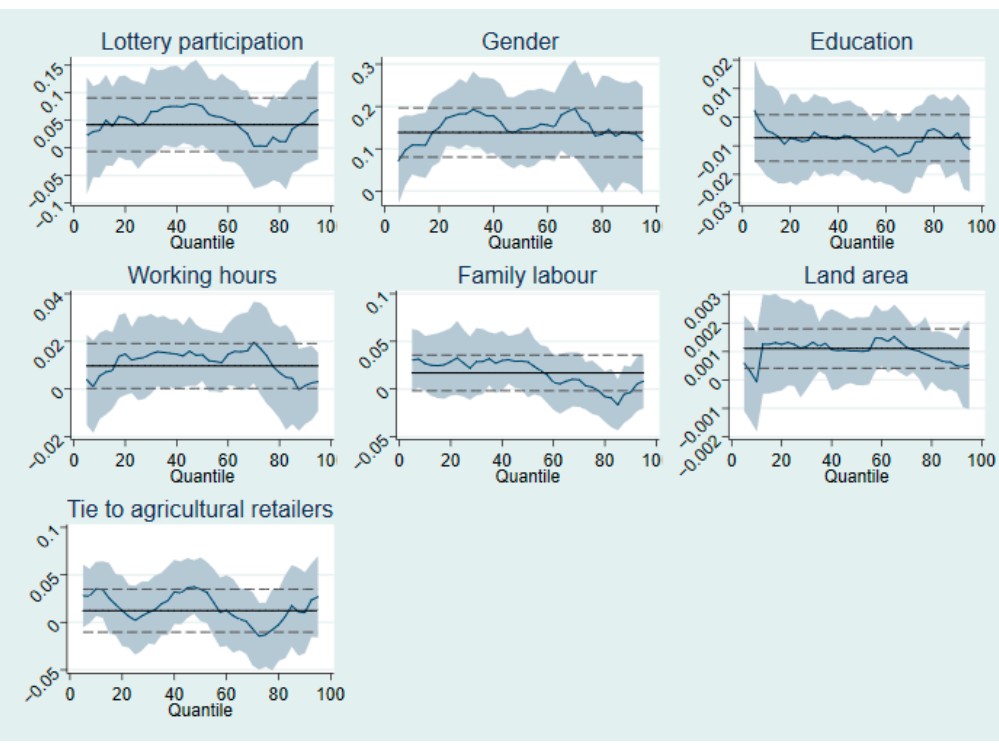

**Figure 2.** The impacts of factors on technical efficiency under different quantiles. Note: Shaded areas represent 95 percent confidence band for the quantile regression estimates. The black solid lines denote the conventional 95 percent confidence intervals for the OLS coefficient.

*4.4. Analysis of Mediating Factors*

Table 8 reports the regression results of Equation (13). Column (1) shows a significant positive effect of farmers' lottery participation on technical efficiency, complying with the above results. Column (2) indicates that risk-seeking farmers are more likely to use biopesticides instead of traditional chemical pesticides. Column (3) shows that farmers who use biopesticides can significantly contribute to the technical efficiency of banana cultivation. Comparing the results in Columns (1) and (3), the coefficient of the effect of lottery participation on agricultural technical efficiency decreases from 0.042 to 0.033 when the mediating variables are added to the model. This suggests that the use of biopesticides plays a significant and complete mediating effect between farmers' lottery participation and technical efficiency. The weight of the mediating effect is 21.49% ($0.094 \times 0.096 / 0.042$), indicating that 21.49% of the impact of lottery participation on farmers' technical efficiency can be explained and mediated by the use of biopesticide. Pest and disease control during agricultural production in developing countries has mainly relied

on chemical pesticides [72]. However, many scholars have confirmed that the excessive use of chemical pesticides would seriously aggravate water and soil pollution and damage human health [65,73]. In recent years, the Chinese local government has introduced the policy of the "double reduction" of pesticides and chemical fertilisers for green agricultural development. Biopesticides are considered a necessary alternative to chemical pesticides and a way to achieve agricultural sustainable development due to their low residue, low toxicity and environmental friendliness [74].

**Table 8.** Results of the mediating effect.

| | Path1 | | | Path2 | | |
|---|---|---|---|---|---|---|
| | (1) PE | (2) Biopesticides | (3) PE | (4) PE | (5) Soil Improvement | (6) PE |
| Lottery participation | 0.042 * (0.025) | 0.094 * (0.047) | 0.033 (0.024) | 0.042 * (0.025) | 0.035 (0.046) | 0.040 * (0.025) |
| Biopesticides | | | 0.096 *** (0.026) | | | |
| Soil improvement | | | | | | 0.053 ** (0.027) |
| Control variables | | Controlled | | | Controlled | |

Note: Standard errors in parentheses. Significance level: *** $p < 0.01$, ** $p < 0.05$, * $p < 0.1$.

Column (5) shows that there is no significant correlation between the banana farmers' lottery participation and the adoption of soil improvement practices, suggesting that the mediating role of the soil amendment practice does not exist. However, the results demonstrate that the soil improvement practice can significantly improve farmers' technical efficiency. Lloret et al. [66] suggested that soil amendments could be used to improve soil structure, reduce salinity hazards, increase infiltration rates and improve water and fertiliser use efficiency, resulting in increased crop yields.

## 5. Conclusions and Policy Implications

This paper assesses the impact of farmers' lottery participation on their technical efficiency using an ESR model that takes into account the sample selection bias and explores the mediating factors through a mediation model. The present study can be concluded with three main findings. First, 59 percent of the farmers in the sample participated in the lottery, and the choice of lottery participation is determined by the farmers' characteristics, including gender, working hours and household labour. Second, the farmers' lottery participation, as a revealed risk preference, has a statistically significant and positive impact on their agricultural technical efficiency. The ATT estimation suggests that the technical efficiency of the farmers who participated in lottery was largely increased (by 41.59%). Furthermore, the quantile regression results suggest that lottery participation has heterogeneous effects on farmers' technical efficiency. The impact of lottery participation on economic performance is more significant for farmers with middle levels of technical efficiency. Third, the adoption of new technologies mediates the impact of lottery participation on technical efficiency. For example, farmers who participate in the lottery have a higher probability of using biopesticides, and therefore improve the technical efficiency of their banana farms. There was no statistically significant difference in the willingness of the two groups of farmers to adopt the soil improvement technique, suggesting that risk preference was not a factor in the adoption of the technology. However, the productivity of banana farmers who adopted the soil improvement technique increased significantly.

The study findings have several important implications. First, policymakers should identify the different types of risk attitudes of farmers when promoting new agricultural technologies. For example, for risk-seeking farmers, the new agricultural technologies with high risks and high returns can be made available to them. Policymakers could highlight the potential benefits of adopting new technologies and provide support to help them manage

the risks, whilst the promotion of agricultural technologies with lower risk can target risk-averse farmers. Of course, policymakers can provide more information and resources to help them better understand the new technologies. Second, the government can provide access to green agricultural technology for smallholder farmers. Smallholder farmers can improve their technical efficiency by using biopesticides or soil improvement techniques. These technologies can bring benefits to the agricultural production and minimise the negative impacts on the environment and human health. Therefore, the government, NGOs and stakeholders may provide smallholder farmers with technical assistance and encourage them to adopt these practices. Third, agricultural extension workers can develop farming training programs that are easy to follow for smallholder farmers, including crop management, soil conservation, and pest control. Such training programs can be delivered through farmer field schools and network platforms. In addition, the technicians can make regular visits to farmers in the field and provide technical assistance to help smallholder farmers improve their technical efficiency in agriculture.

Although this study makes a marginal contribution to the existing literature on the impact of farmers' risk preferences on their technical efficiency, it still has limitations. First, if some farmers become accustomed to participating in the lottery in their daily life, lottery participation can be a habit for them, and may not reflect their risk attitudes to some extent. Thus, the frequency of lottery participation may be an important variable that can be considered in future studies. Second, although the ESR model was used to address the issue of biased estimation, this study did not take into account some other possible factors that may influence farmers' adoption of new technologies, such as input subsidies and credit availability. This can be enhanced in future studies.

**Author Contributions:** M.W.: conceptualisation, methodology, investigation, formal analysis, and writing—original draft. Y.Z.: conceptualisation, investigation, formal analysis, project administration, funding acquisition, and writing—review and editing. All authors have read and agreed to the published version of the manuscript.

**Funding:** This work was supported by the Hainan Provincial Natural Science Foundation of China (No. 720RC581), the National Natural Science Foundation of China (No. 71863006), and the China Agriculture Research System of MOF and MARA (No. CARS-31).

**Institutional Review Board Statement:** Not applicable.

**Data Availability Statement:** The raw data supporting the conclusions of this article will be made available by the authors, without undue reservation.

**Conflicts of Interest:** The authors declare no conflict of interest.

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
