# Peer review of "Does Farmers’ Lottery Participation Affect Technical Efficiency of Banana Production in Rural China?"

_agriculture, doi:10.3390/agriculture13040767_

Round 1
Reviewer 1 Report
The work is well presented.
The introduction can be made slightly short and the authors can think of making a separate section on literature review.
The policy implications can be reviewed as they look very generic.
The paper addresses a very prominent contemporary issue, i.e., agricultural production efficiency. The topic is relevant.
The authors' usage of participation in a lottery by farmers to access their risk appetite makes the treatment of the topic interesting.
The econometric model used is appropriate. The Endogenous switching regression (ESR) and the mediation model have been appropriately used by the authors, and the results have been well-presented.
The findings of this study are consistent and in line with existing studies. Risk-averse farmers are less likely to adopt new technology and invest less in technology resulting in lower productivity.
However, it is important to understand that there are other aspects or factors which influence the decision of a farmer household in adopting new technology such as - production contract, input subsidy, credit availability, proper training and knowledge about its benefits/ consequences.
If these factors are taken into account, the risk appetite of the farmers and their will to adopt new technology can change and may not be linked with their willingness to participate in a lottery.
Here is an interesting study related to risk preference.
Arslan, R. C., Brümmer, M., Dohmen, T., Drewelies, J., Hertwig, R., & Wagner, G. G. (2020). How people know their risk preference. Scientific reports, 10(1), 15365.
Authors can take a look at this paper.
Author Response
Revision — authors’ response:
Dear Editor and Reviewers,
We appreciate the valuable comments from reviewers and editors. These comments have been carefully taken into consideration for preparing our revision, and we have addressed the issues raised by the reviewers. The point-to-point responses are given as below:
Response to Reviewer 1 Comments:
Point 1: The work is well presented. The introduction can be made slightly short and the authors can think of making a separate section on literature review. The policy implications can be reviewed as they look very generic.
Response 1: We sincerely thank you for your valuable comment concerning our manuscript. As you know, it is also common to incorporate the literature review into the introduction Section. For example, the following papers that have been published in this journal:
- Olasehinde, T.S.; Qiao, F.; Mao, S. Impact of improved maize varieties on production efficiency in nigeria: separating technology from managerial gaps. Agriculture 2023, 13, 611. doi:10.3390/agriculture13030611.
- Xing, X.; Zhang, Q.; Ye, A.; Zeng, G. Mechanism and empirical test of the impact of consumption upgrading on agricultural green total factor productivity in china. Agriculture 2023, 13, 151. doi:10.3390/agriculture13010151.
However, we added more literature which is associated with the present study and enhanced literature review part in the revision. (See lines 59-75)
We have reviewed the policy implications to make it more specific and relevant to our conclusion. The revised part is given as below: (See lines 466-484)
[The study findings have several important implications. First, policymakers should identify the different types of risk attitudes of farmers when promoting new agricultural technologies. For example, for risk-seeking farmers, the new agricultural technologies with high risks and high returns can be made available to them. Policymakers could highlight the potential benefits of adopting new technologies and provide support to help them manage the risks. Whilst, the promotion of agricultural technologies with lower risk can target risk-averse farmers. Of course, policymakers can provide more information and re-sources to help them better understand the new technologies. Second, the government can provide access to green agricultural technology for smallholder farmers. Smallholder farmers can improve their technical efficiency by using biopesticides or soil improvement techniques. These technologies can bring benefits to the agricultural production, and minimise negative impacts on the environment and human health. Therefore, the government, NGOs, and stakeholders may provide smallholder farmers with technical assistance and encourage them to adopt these practices. Third, agricultural extension workers can develop farming training programs that are easy to follow for smallholder farmers, including crop management, soil conservation, and pest control. Such training programs can be delivered through farmer field schools and network platforms. Also, the technicians can make regular visits to farmers in the field, and timely provide technical assistance to help smallholder farmers improve technical efficiency in agriculture.]
Point 2: However, it is important to understand that there are other aspects or factors which influence the decision of a farmer household in adopting new technology such as - production contract, input subsidy, credit availability, proper training and knowledge about its benefits/ consequences.
If these factors are taken into account, the risk appetite of the farmers and their will to adopt new technology can change and may not be linked with their willingness to participate in a lottery.
Response 2: We agree with your concern about possible omitted variables that may lead to biased estimation results. To tackle this potential issue, a) we collected the first-hand data of farmers through a questionnaire, in which we considered variables as possible as we can according to the context of study area; b) we also followed A multistage sampling method to sample the farmers, and make sure that respondents are randomly selected; and c) we used an endogenous switching regressions (ESR) model to avoid biased estimation due to the endogeneity problem. It is an important strategy to make our results reliable.
However, we still added it as one of limitations of our study at the end of the manuscript. (See lines 485-493)
[Although this study makes a marginal contribution to the existing literature on the impact of farmers’ risk preferences on technical efficiency, it still has limitations. First, if some farmers become accustomed to buy the lottery in their daily life, lottery participation can be a habit of them, and may not reflect their risk attitudes to some extent. Thus, the frequency of lottery participation may be an important variable that can be considered in future study. Second, though ESR model was used to address the issue of biased estimation, this study did not take into account some other possible factors that may influence farmers’ adoption of new technologies, such as input subsidies, and credit availability. It can be enhanced in future study.]
Point 3: Here is an interesting study related to risk preference.
Arslan, R. C., Brümmer, M., Dohmen, T., Drewelies, J., Hertwig, R., & Wagner, G. G. (2020). How people know their risk preference. Scientific reports, 10(1), 15365.
Authors can take a look at this paper.
Response 3: We appreciate the reviewer's recommendation. This is an interesting literature and related to our study. We have cited it and highlighted in red color. (See line 82)
Thank you so much for your comments and suggestions!
Authors

Reviewer 2 Report
The title must be reviewed, because it is unclear and although the manuscript is sufficiently informative in the impact of lottery participation on the production efficiency of banana farms, but it must be revised in the light of the implications of this issue instead of agricultural production. It is difficult to figure out the banana farmers in rural China as agricultural production farmers. It is important to specify how the efficiency of banana farmers’ participation in lottery have guided this research, and how approaches related to these different views were used to test the hypothesis.
Regarding the methodology, the authors are suggested to use an theoretical approach to support the variables included in the Endogenous Switching Regression (ESR) model, in the efficiency of Farmers’ lottery participation model (lines 271-272), and in the mediation model (line 216). In the same way, the authors must justify the importance of the Hainan province and the four counties as banana-growing towns. In this sense, it could be more convenient to use analysis techniques that explain how much variance is due to banana-growing towns and how much to farms.
The content of the Results and Discussion must be in terms of the efficiency of banana Farmers’ lottery participation model, but not in agricultural production efficiency. The conclusion does not summarize the findings; Issues such as “cleaner production technologies (lines 448), adoption of new agricultural technologies (lines 449), food security and the sustainable development (lines 450), an the impact of farmers’ lottery participation on agricultural production efficiency (lines 451-452) were not studied.
Specific comments.
Line 459: lottery participants' production efficiency?
Line 473: importance of environmental sustainability? How was it estimated?
Line 473-474: dissemination of green agricultural technologies? How was it estimated?
Author Response
Revision — authors’ response:
Dear Editor and Reviewers,
We appreciate the valuable comments from reviewers and editors. These comments have been carefully taken into consideration for preparing our revision, and we have addressed the issues raised by the reviewers. The point-to-point responses are given as below:
Response to Reviewer 2 Comments:
Point 1: The title must be reviewed, because it is unclear and although the manuscript is sufficiently informative in the impact of lottery participation on the production efficiency of banana farms, but it must be revised in the light of the implications of this issue instead of agricultural production. It is difficult to figure out the banana farmers in rural China as agricultural production farmers. It is important to specify how the efficiency of banana farmers’ participation in lottery have guided this research, and how approaches related to these different views were used to test the hypothesis.
Response 1: Thank you so much for your suggestion. We agree with reviewers’ that the previous title is somehow confusing. In fact, this work focuses on the impact of farmers’ lottery participation on technical efficiency of their farms. Agricultural production is a broad term and not to the point of this study directly, thus, we change the title into: Does farmers’ lottery participation affect technical efficiency of banana production in rural China? (See line 2-3 in Page 1) We believe the revised title is more concise and accurate. Besides, we revised all relevant expressions about technical efficiency in the manuscript. According to the comment, we also enhanced the introduction section to better introduce the research objectives.
Point 2: Regarding the methodology, the authors are suggested to use an theoretical approach to support the variables included in the Endogenous Switching Regression (ESR) model, in the efficiency of Farmers’ lottery participation model (lines 271-272), and in the mediation model (line 216). In the same way, the authors must justify the importance of the Hainan province and the four counties as banana-growing towns. In this sense, it could be more convenient to use analysis techniques that explain how much variance is due to banana-growing towns and how much to farms.
Response 2: Thanks for your comments and suggestion. We have used the existing literature to support the variables included in the ESR model and the mediation model. (See line 133-135 in Page 3 and line 213-214 in Page 5) Likewise, we added the literature to explain the setting of the mediation model. (See line 212-215 in Page 5)
[The theoretical basis of the mediation model is rooted in social psychology, where researchers often seek to understand the mechanisms on how one factor influences another [29]. The mediation model makes the assumption that the mediator partially or completely explains the link between the independent and dependent variable. To verify the mediating role of technology adoption between banana farmers’ lottery participation and technical efficiency of their farms, this study follows a stepwise regression method from Baron and Kenny [52].]
According to the comment, we justified the importance of Hainan Province and these four counties as banana growing towns using data. (See line 237-247 in Page 6)
[According to the China National Banana Industry Technology System (CNBITS), the total production of banana in China was 5.348 million tons in 2020, while Hainan province ranks as the third largest production area in China with a total production of 1.084 million tons…
According to the Hainan Provincial Bureau of Statistics, the area of bananas harvested in the aforementioned four counties accounted for 63.5% of the province's total in 2019, therefore the four sample counties are of considerable relevance and representativeness.]
Thanks for the reviewer’s comment. The term “town” may be misleading for reviewers. In the study area, a town actually consists of serval villages. Banana farmers in the sample villages belong to the town, too. Therefore, we added the statement about the context of study area to make it clear to international readers. (See lines 247-251 in Page 6)
[two to four towns were randomly selected from each sample county. In the study area, a town actually consists of serval villages. And three to five villages were randomly selected from each town.]
Point 3: The content of the Results and Discussion must be in terms of the efficiency of banana Farmers’ lottery participation model, but not in agricultural production efficiency. The conclusion does not summarize the findings; Issues such as “cleaner production technologies (lines 448), adoption of new agricultural technologies (lines 449), food security and the sustainable development (lines 450), an the impact of farmers’ lottery participation on agricultural production efficiency (lines 451-452) were not studied.
Response 3: Thanks for reviewer’s valuable comments. We revised the title of the manuscript and replaced "Agricultural Production Efficiency" with "Technical Efficiency" which is more accurate and in line with our study. Following the comment, we removed the irrelevant content and presented the findings correctly in the conclusion section. Please check the lines 447-465.
[This paper assesses the impact of farmers’ lottery participation on technical efficiency using an ESR model that takes into account sample selection bias, and exploring the mediating factors through a mediation model. The present study can be concluded with three main findings. First, there are 59 percent of the farmers in the sample participated in the lottery. And the choice of lottery participation is determined by the farmers’ characteristics, including gender, working hours and household labour. Second, farmers’ lottery participation, as a revealed risk preference, has a statistically significant and positive impact on agricultural technical efficiency. The ATT estimation suggests that technical efficiency of farmers who participated in lottery was largely increased (by 41.59%). Furthermore, the quantile regression results suggest that lottery participation has heterogeneous effects on farmers’ technical efficiency. The impact of lottery participation on economic performance is more significant for farmers with middle levels of technical efficiency. Third, the adoption of new technologies mediates the impact of lottery participation on technical efficiency. For example, farmers with lottery participation have higher probability to use biopesticides, and therefore improve the technical efficiency of their banana farms. There was no statistically significant difference in the willingness of the two groups of farmers to adopt the soil improvement technique, suggesting that risk preference was not a factor in the adoption of the technology. However, the productivity of banana farmers who adopted the soil improvement technique increased significantly.]
Point 4: Line 459: lottery participants' production efficiency?
Response 4: Thanks so much for the comment. It should be “technical efficiency of farmers who participated in lottery”, and we have revised it. (See line 454-455 in Page 13)
Point 5: Line 473: importance of environmental sustainability? How was it estimated?
Line 473-474: dissemination of green agricultural technologies? How was it estimated?
Response 5: Thanks so much for the comment. In fact, our study did not estimate the importance of environmental sustainability and the dissemination of green agricultural technologies. we have removed content that is not relevant to this study.
Thank you so much for your comments and suggestions!
Authors

Reviewer 3 Report
Look at the annotations in the manuscript

Author Response
Revision — authors’ response:
Dear Editor and Reviewers,
We appreciate the valuable comments from reviewers and editors. These comments have been carefully taken into consideration for preparing our revision, and we have addressed the issues raised by the reviewers. The point-to-point responses are given as below:
Response to Reviewer 3 Comments:
Point 1: The manuscript "Impacts of farmers’ lottery participation on agricultural production efficiency in rural China: An analysis based on multiple applied econometric models" enters the topics of the journal. In this study, the utility is verified.
The objective of the work was to evaluate the individual's risk preference of farmer on banana production in China, through lottery participation. This work makes two interesting contributions. On the one hand, it shows the sociodemographic characteristics associated with the perception of risk in the farmers of China. On the other hand, it highlights the value of simple tools, such as participation in the lottery, to assess attitudes towards risk and its relationship with levels of efficiency.
Response 1: Thanks so much for the comment.
Point 2: In the introduction it is recommended to delve deeper into the Theory of Planned Behavior
(TPB), the attitude towards risk, the theory of expected utility, etc.
Response 2: Thanks to the reviewer's suggestion, we have added the content about Theory of Planned Behavior and other theoretical basis related to this study in the introduction, and highlighted in red color. (See line 59-75 in Page 2)
[Agricultural production of small-scale farms in developing countries is highly vulnerable to natural disasters and market price fluctuations [16,17]. The natural and market risks threaten the livelihood of smallholder farmers. In existing studies, the theory of planned behaviour (TPB) has been widely used to understand farmers intentions or be-haviour in agricultural production [18,19]. Under risks and uncertanties, smallholder farmers often make decisions upon their prior farming experiences. In order to effectively prevent shocks to livelihoods and stabilise crop yields, farmers could adopt ex-ante comprehensive management strategies, such as crop diversification, agricultural insurance, etc.[20–22]. However, it cannot be overlooked that the adoption of coping strategies is in-separable from the attitudes of smallholders towards risks. Risk preference is farmers’ attitudes and decision-making behaviour under risks in the agricultural production. According to previous studies, farmers’ risk preferences have three categories: risk-seeking, risk-neutral and risk-averse [23–25]. The existing literature also suggests that the adoption of new technologies is often associated with farmers’ risk preferences [26,27]. Risk-averse farmers are more likely to follow traditional practices and technologies, while risk-seeking farmers are more likely to embrace new crop varieties and technologies and benefit from them [24,28,29].]
Point 3: * Eliminate paragraphs that escape the objective of the work. Line 55-64;
Response 3: We deleted line 55-64. Thanks so much for the comment.
Point 4: * Move paragraph 89-94, which corresponds to results or discussion
Response 4: Thanks so much for the comment. We moved this part to the objectives of this study. (See line 90-105 in Page 2-3)
[This work has the following two research objectives. The primary objective of this study is to examine the impact of farmers’ lottery participation on technical efficiency. Generally, lottery participation can reveal farmers’ risk preferences. risk-seeking farmers may be more inclined to use new technologies or practices to increase the technical efficiency. However, it also potentially have a negative impact on agricultural production. For example, farmers may spend too much time and energy on the lottery, and it may reduce their inputs in agriculture. Thus, it is worth an empirical exploration. The second objective of this research is to detect the mediating factors between the two variables, if farmers’ lottery participation affects their technical efficiency. This study suggests two possible pathways. The first path is that farmers are addicted to lotteries, reporting a passive attitude towards farming, thus inhibiting technical efficiency. Another path is that farmers with lottery participation are risk-seekers, and prefer to adopt new technologies in agriculture, thereby increasing the technical efficiency. Thus, the adoption of new technology can be a reasonable mediating factor between farmers’ lottery participation and agricultural technical efficiency. This paper tests it using a mediation model based on the data collected from banana farmers in rural China.]
Point 5: Remove line 104-105. reasoning too simple.
Response 5: We removed line 104-105. Thanks so much for the comment.
Point 6: The title should be modified according to the objective. I propose:“Assessment of individual's risk preference of farmer on banana production in China, through lottery participation” or similar.
Response 6: Thanks so much for your comments and suggestion. The objective of this study is to empirically investigate the impact of farmers’ lottery participation on technical efficiency in banana production, and identify the mediating factors between the two variables. After the cautious consideration, we changed the title as: Does farmers’ lottery participation affect technical efficiency of banana production in rural China? (See line 2-3 in Page 1)
Point 7: There are several ways to approach the attitude to risk. I will give you a summary of the works found
*Qualitative assessment (likert scale). Different authors propose that assessing the subjective attitude towards risk using a scale in which the subject reported her attitude towards risk, where 1 represented the maximum aversion and 10 the maximum propensity to risk.
* Quantitative valuation. Through two games of chance.
"Objective" risk attitude was measured by two lottery games, adapted from the German Socio-Economic Panel Study (SOEP). Frequently these games reproduce television contests in which the participant chose a box between two possible ones. One of them had a prize and the other was empty. The contestant received the contents of the chosen box. Simultaneously, he received economic "offers" that she could accept in exchange for not participating in the contest. The offers were increased if the contestant declined and chose to remain in the contest, until the contestant either accepted the amount of money offered and withdrew from the contest, or declined the highest offer and chose to play regardless.
This path has two variants or different games:
- The first without the possibility of economic losses and offered a maximum profit
- The second in which to play it was necessary to pay an amount that could be lost and whose maximum profit was 4 or 5 times the amount.
In this case, you choose to simply ask if you participate in the lottery or not. Explain why you use this methodology, advantages and disadvantages. Wouldn't a combination of the
above have been more appropriate?
Response 7: Thanks so much for the valuable comments. Both the qualitative and quantitative methods mentioned by the reviewers are very important to our study, we have cited related work in our study (see line 78-84 in Page 2). The method used to measure individual risk attitudes differs from previous studies. In the context of our study, farmers’ lottery participation is actual action that can directly observed and recorded in the reality without any intervention from observers. In this way, the collected data is not distorted by any unobservable factors. Comparatively, the designed experiments may not accurately reflect the risk preference of participants. Therefore, this study uses lottery participation as a proxy for risk preferences of farmers. However, if some farmers become accustomed to buy the lottery in their daily life, lottery participation can be a habit of them, and may not truly reflect their risk attitudes as time goes by. This might be the disadvantage. We added it as one of the limitations of this work at the end of the manuscript. (See line 485-493 in Page 14)
[Although this study makes a marginal contribution to the existing literature on the impact of farmers’ risk preferences on technical efficiency, it still has limitations. First, if some farmers become accustomed to buy the lottery in their daily life, lottery participation can be a habit of them, and may not reflect their risk attitudes to some extent. Thus, the frequency of lottery participation may be an important variable that can be considered in future study. Second, though ESR model was used to address the issue of biased estimation, this study did not take into account some other possible factors that may influence farmers’ adoption of new technologies, such as input subsidies, and credit availability. It can be enhanced in future study.]
Point 8: In table 4, remove the Diff column and simply show the P-values.
Response 8: We removed the Diff column and show the P-values. (See table 4)
Point 9: In the discussion explain the differences between those who participated and those who did not. I think it is important, because the control group already shows important differences.
I think the reasoning is as follows: There is a series of sociodemographic factors (gender, Education, working hours, family labor, size, tie to agricultural retailers among others) that conditions the attitude towards risk. This attitude is visualized in the participation in the
lottery that is associated with the degree of innovation and efficiency.
On the other hand, the population behaves heterogeneously by quintiles. It would not be more convenient to make a previous grouping or typology of farms (multivariate techniques) and then in each group see its relationship with participation in the lottery.
Response 9: Thanks so much for the valuable comments. We agree with reviewers that it is very important to explain the differences between those who participated and those who did not in the discussion. In our work, the first stage of the ESR model presents factors that may lead farmers to participate or not participate in the lottery. We explained the influence of sociodemographic factors on farmers’ lottery participation which reveals farmers’ risk preferences, and also discussed results of each variable in contrast to existing studies. (See line 323-345 in Page 9-10). The quantile regression model was used to analyse the heterogeneous effects under different levels of farmers’ technical efficiency. Because this study has two objectives, we did not give more discussion solely on farmers’ lottery participation. But, the comment is still useful for us to consider in future study.
[The first stage of ESR model estimated the determinants of farmers’ lottery participation, and the results are presented in the third column of Table 5. Male farmers, smartphone, and family members' attitude towards the lottery showed a statistically significant and positively associated with farmers’ lottery participation. Specifically, men are more inclined to participated in lotteries than women, a finding that is consistent with the findings in previous studies [58]. Generally, men have a stronger preference of adventure than women. Smartphones usage can promote farmers’ lottery participation. The finding is somehow consistent with Törrönen et al. [59]. The coefficient of attitude of family members towards lottery is positive and significant, suggesting that if family members have a supportive attitude towards the lottery, the more likely the farmer participated in it. Existing evidence also showed that children are involved when their parents are addicted to lotteries [60]. It should be noted that the selection equation in ESR model accounts for un-observed heterogeneity that could bias the treatment effect of lottery participation on technical efficiency. For this reason, the selection equation needs to include a valid instrument, which should be excluded in the outcome equation [44]. In contrast, working hours, family labour, and tie to agricultural retailers negatively influenced farmers’ lottery participation. Picchio et al. [61] suggested that winning the lottery prize could reduce people’s working hours, while our study shows that the probability of farmers’ lottery participation decreases significantly as their average working hours and family labour increase. Agricultural retailers are the important node for transmitting the market information among farmers in rural China. The tie to agricultural retailers shows a negative and statistically significant association with farmers’ lottery participation, suggesting that farmers with closer interaction with agricultural retailers are less likely to buy the lottery.]
Point 10: Remove from 448 to 455 lines.
Response 10: We removed line 448-455. Thanks so much for the comment.
Point 11: The study identifies, through participation in the lottery, different social and structural factors that could be acted upon to improve efficiency. In quintile 1...identify, in quintile 2, identify... etc. It is a good job and I encourage its correction. But you have to focus the work according to the objective. I think that this paper is very promising.
Response 11: Thanks for the comment. We have re-structure the introduction part to clearly present the research objectives. According to the valuable comments, we changed the title of this manuscript. We removed irrelevant expressions and enhanced the manuscript to better follow the objectives.
Once again, thank you very much for your comments and suggestions!
Authors

Round 2
Reviewer 2 Report
Comments and sugestion have been fully addressed and the paper has been revised accordingly